# Atomic scale crystal field mapping of polar vortices in oxide superlattices

Sandhya Susarla [1,2,3✉], Pablo García-Fernández [4], Colin Ophus [1], Sujit Das[2], Pablo Aguado-Puente [5], Margaret McCarter[2,3], Peter Ercius [1], Lane W. Martin [2,3], Ramamoorthy Ramesh [1,2,3,6✉] & Javier Junquera [4✉]

Polar vortices in oxide superlattices exhibit complex polarization topologies. Using a combination of electron energy loss near-edge structure analysis, crystal field multiplet theory, and first-principles calculations, we probe the electronic structure within such polar vortices in $[(PbTiO_3)_{16}/(SrTiO_3)_{16}]$ superlattices at the atomic scale. The peaks in Ti $L$-edge spectra shift systematically depending on the position of the $Ti^{4+}$ cations within the vortices i.e., the direction and magnitude of the local dipole. First-principles computation of the local projected density of states on the Ti $3d$ orbitals, together with the simulated crystal field multiplet spectra derived from first principles are in good agreement with the experiments.

[1] National Center for Electron Microscopy, Molecular Foundry, Lawrence Berkeley National Laboratory, Berkeley, CA 94720, USA. [2] Department of Materials Science and Engineering, University of California, Berkeley, CA 94720, USA. [3] Materials Sciences Division, Lawrence Berkeley National Laboratory, Berkeley, CA 94720, USA. [4] Departamento de Ciencias de la Tierra y Física de la Materia Condensada, Universidad de Cantabria, Cantabria Campus Internacional, Avenida de los Castros s/n, 39005 Santander, Spain. [5] CIC nanoGUNE BRTA, Donostia - San Sebastián 20018, Spain. [6] Department of Physics, University of California, Berkeley, CA 94720, USA. ✉email: ssusarla@lbl.gov; rramesh@berkeley.edu; javier.junquera@unican.es

Epitaxial complex oxide heterostructures and superlattices with their interplay of spin, charge, orbital and lattice degrees of freedom offer a rich platform to study exotic phenomena such as spin-charge transfer, multiferroicity, and unique topological phases[1]. With careful manipulation of the elastic, electrostatic and gradient energies, topological structures such as polar flux-closure domains[2,3], vortices[4–7], bubble domains[8,9] and skyrmions[10,11], can be formed in epitaxially grown $(PbTiO_3)_n/(SrTiO_3)_n$ (PTO/STO) superlattices and other ferroelectric nanocomposites[12]. In particular, polar vortices, i.e., smoothly rotating electric dipoles, are interesting for their spatially confined negative capacitance[13] and chirality[14]. The structure and dipole arrangement in polar vortices has been studied via X-ray scattering techniques, (scanning) transmission electron microscopy ((S)/TEM), phase field simulations, and atomistic first- and second-principles calculations[4,7]. The effect of external stimuli such as electric fields[15,16] or mechanical stress[17] on the topological transformations have also been examined.

However, the fundamental correlation between the atomic structure and the electronic structure (which is manifested in the chemical bonding) has heretofore not been explored. The hybridization between nominally empty $d$ orbitals on the B-site with the occupied O $2p$ orbitals favors the condensation of a polar (ferroelectric) state in $ABO_3$ perovskite oxides[18]. The complex, continuously rotating local polarization texture of the vortices, in turn, can result in especially intricate $d$-orbital interactions. There are only a few reports in the literature discussing these orbital interactions within vortices[14,19] primarily using resonant soft X-ray diffraction and spectroscopy. Although soft X-ray spectroscopy can probe these interactions at the transition metal $L$-edge, these techniques do not have the spatial resolution to resolve variations within one vortex (~5 nm region). Electron energy loss spectroscopy (EELS) in the STEM mode uses inelastically scattered electrons to probe the core-shell excitations (empty density of states) of transition metals at atomic resolution[20–25].

We studied the crystal field of the Ti $L$-edge in polar vortices formed in $[(PbTiO_3)_{16}/(SrTiO_3)_{16}]_8$ superlattices with a combination of high-resolution STEM-EELS mapping using a state-of-the-art direct electron detector and spectrometer (Gatan Continuum with a K3 detector), first-principles calculations, and crystal field multiplet theory. Changes in the crystal field of the $Ti^{4+}$ cations in the PTO/STO superlattices are mapped as the spontaneous displacement of $Ti^{4+}$ (and its corresponding $3d$ orbitals) rotates within the vortices. In doing so, we answer three important questions: (i) How are the $t_{2g}$ and $e_g$ orbitals affected by the local polarization and tetragonality? (ii) How does the rotation of the Ti $3d$-orbitals affect the local crystal field? and, (iii) What is the crystal field at the vortex core, a special region where exotic effects such as local negative capacitance[13] have been reported?

## Results

**Ti polarization orientation in polar vortices.** The PTO/STO superlattices were grown epitaxially on single crystal $[110]_o$ DyScO$_3$ (DSO) by pulsed-laser deposition (PLD) (Methods section). Figure 1a shows a cross-section, high-resolution high-angle annular dark field (HAADF-) STEM image of a 100 nm $[(PTO)_{16}/(STO)_{16}]_8$ superlattice. The difference in the atomic number between Pb ($Z = 82$) and Sr ($Z = 38$) cations allows for the identification of the two layers with an atomically sharp interface between them. The brighter layers correspond to PTO and the darker layers to STO[26]. Figure 1b shows the corresponding low angle annular dark field (LAADF)-STEM image which display the periodic local strain fields within the PTO layer[4]. To precisely determine the location of polar

vortices, we measure the displacement of $A$-site cations (Pb/Sr atoms) in the high-resolution HAADF-STEM image (white box in Fig. 1a, b) using drift correction and Gaussian fitting of each $A$-site[27] (Methods section and Supplementary Fig. 1). Figure 1c shows both the magnitude (arrow length) and direction (arrow direction) of the displacement of the $A$-site cations with respect to the centrosymmetric positions overlaid on the HAADF-STEM images. Figure 1c also shows a non-zero curl of the displacement vectors as deep red/blue contrast at the center of the PTO layer.

**Electronic structure description of the Ti $3d$ orbitals in vortices.** With the atomic structure described, we turn our attention to the electronic structure. In a purely ionic model, the formal charge of Ti would be +4 in both the PTO and STO, with an empty $d$ orbital. At room temperature, STO adopts a $Pm\bar{3}m$ cubic perovskite structure ($c/a = 1$), with the $Ti^{4+}$ cation in a high-symmetry position (i.e., no displacement) at the center of the oxygen octahedra. The five-fold degenerate atomic $d$ level of the $Ti^{4+}$ splits into lower energy three-fold $t_{2g}$ and higher energy two-fold $e_g$ levels in the resulting octahedral $O_h$ crystal field. Tetragonality ($c/a$) and local polarization can further cause an internal splitting within the $e_g$ and $t_{2g}$ levels. First, the epitaxial strain imposed by the DSO substrate, ($a_{pc} = 3.952$Å where $pc$ stands for pseudo-cubic) translates into tensile strains of +1.20 % on STO, and +1.30 % on PTO, where the epitaxial strain has been computed with respect to the cubic lattice parameters of SrTiO$_3$ and the tetragonal phase of PbTiO$_3$ at room temperature. Due to the combination of the epitaxial constraint and spontaneous polarization, the $Ti^{4+}$ cations move off-center and lift the degeneracy of the $e_g$ and $t_{2g}$ levels. The electrostatic boundary conditions imposed on the superlattice leads to a continuous rotation of the polarization and the formation of vortices in which the local crystal field ($e_g$ and $t_{2g}$ splitting) in the $TiO_6^{2-}$ octahedra changes depending on its position within the vortex (Fig. 1d). These subtle changes can be detected with EELS[28–30].

**EEL spectroscopy and mapping of Ti $L$ edge.** We performed near-edge core loss STEM-EELS mapping with a monochromated electron beam to understand the subtle electronic changes within the vortices. Figure 2a shows the simultaneous HAADF-STEM image that was acquired along with EELS maps. Figure 2b shows the corresponding A site displacement vector maps where the white circles indicate the location of the vortex cores. Figure 2c, d display the average non-negative matrix factorization (NMF) de-noised O $K$ and Ti $L$ edge EEL spectra of the STO (green), PTO vortex edge (orange), and PTO vortex core (blue). The details of the de-noising process are reported in the Methods section. Each colored square in Fig. 2a encloses $15 \times 15$ scan positions which were summed to improve the SNR of the spectra shown in Fig. 2c–e. The raw data shows the same features as can be seen in Supplementary Fig. 2.

The O $K$ and Ti $L$ edges provide insights into key electronic properties of the oxide. The O $K$ edge provides information about the bonding of O with the neighboring cations[31]. For example, we can distinguish PTO and STO via the peak indicated by an asterisk (*) in Fig. 2c which is associated with hybridization of O $2p$ levels with Pb $6sp$/Sr $4d$ levels. The first ~5 eV above the onset is related to hybridization between the O $2p$ levels and Ti $3d$ levels. However, since both $\pi$ ($t_{2g}$ levels) and $\sigma$ ($e_g$ levels) interactions are present in such a small energy window, it is difficult to experimentally distinguish subtle changes in hybridization of the Ti-O bonds from the O K edge especially within the vortices in the PTO layer. Thus, the Ti $L$ edge (which arises from

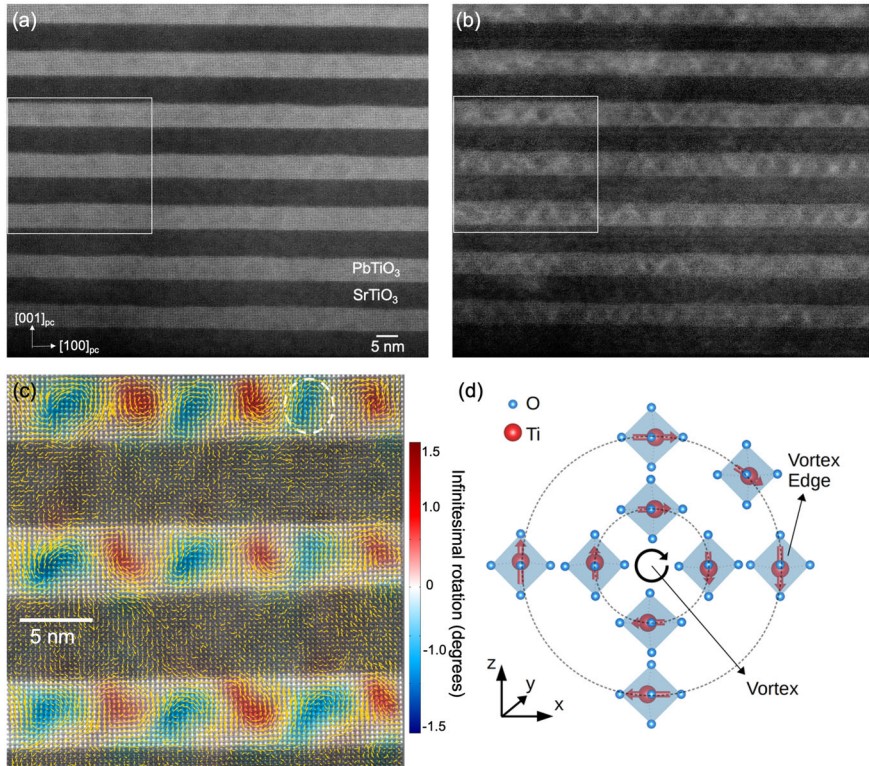

**Fig. 1 Structure and Ti polarization orientation in polar vortices. a** Cross-sectional HAADF- and **b** LAADF-STEM images of $PbTiO_3/SrTiO_3$ superlattices. **c** $A$-site ($A$: Pb or Sr) displacement vectors (yellow arrow) and curl of displacement (red/blue color) overlaid on the HAADF-STEM image. The color bar indicates the magnitude of the curl of the displacement vector. **d** Schematic representing the rotation of $TiO_6$ octahedra within one vortex domain.

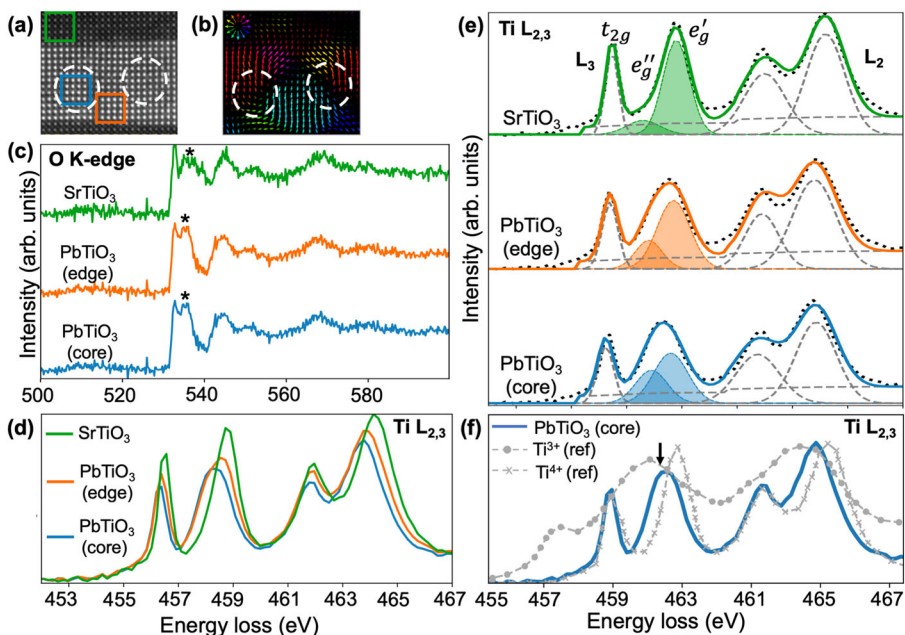

**Fig. 2 EELS of the polar vortices. a** Simultaneously acquired HAADF-STEM image. **b** The corresponding A site displacement vector map with white circles indicating the position of the vortex cores. **c** De-noised O $K$ edge, and **d** Ti $L$ edge spectra of STO, PTO at the vortex core, and edge summed from the green, blue, and orange boxes (respectively) overlaid on **a**. Each box contains 15 × 15 scan positions. **e** Gaussian fits of the Ti $L$ edge EEL spectra with Hartree-Slater background. **f** PTO vortex core spectrum in comparison to $Ti^{3+}$ and $Ti^{4+}$ reference spectra. The absence of any extra broadening in the $t_{2g}$ as compared to the standard $Ti^{4+}$ spectrum negates the possibility of $Ti^{3+}$ in the system.

$2p - 3d$ transitions and has large separation between the $t_{2g}$ and $e_g$ peaks), shows the effects related to subtle changes in hybridization of Ti $3d$ and O $2p$ better. The summed spectra from positions within the colored squares in Fig. 2a are displayed in Fig. 2d for the vortex edge, core and STO, respectively. We observe a negative shift in energy loss for mainly the $e_g$ peak at the vortex core as compared to the vortex edge. These shifts are indicative of either a change in crystal-field, or a local change in the Ti oxidation state[16,28]. Previous EELS studies related to polar vortices have attributed this effect to the presence of $Ti^{3+} - V_{O}$ pairs at the core[16]. To explore this possibility, we compared the experimental vortex core EEL spectra to standard experimental Ti-$L$ edge reference spectra of pure $Ti^{4+}$ (taken from $SrTiO_3$) and pure $Ti^{3+}$ (taken from $YTiO_3$), shown in Fig. 2f[32,33]. A small amount of $Ti^{3+}$ present in a $Ti^{4+}$ system can cause broadening of both the $e_g$ and $t_{2g}$ peaks of the overall spectrum[32,33]. When we compare the vortex core spectrum to standard $Ti^{4+}$, we observe an extra broadening only in the $e_g$ peak indicated by a black arrow in Fig. 2f, and we do not observe any significant change in the $t_{2g}$ peak. The absence of any extra feature on the $t_{2g}$ Ti $L$ edge spectrum at the vortex core as compared to standard $Ti^{4+}$,[32,33] indicates a negligible possibility of the presence of $Ti^{3+}$ in our samples. Our average EEL spectra result also matched with the ensemble XAS spectra acquired from the same material, (Supplementary Fig. 3) eliminating the possibility that the extra $e_g$ broadening and negative shift arises from electron beam induced damage or TEM sample preparation damage. We can thus ascribe the changes in the spectra, in particular the negative shift of the $e_g$ peak, to the crystal-field change within a $Ti^{4+}$ oxidation state framework.

As discussed above, the combination of both strain and polarization produces a deformation of the shape of the $e_g$ peak. We separated this into two contributions: a higher-energy peak ($e_g'$) and a lower-energy peak ($e_g''$). To understand this further, we fitted the Ti $L$ edge at each STEM scan position with Gaussians (one for the mostly symmetric $t_{2g}$ peak and two for the previously mentioned $e_g$ peak) after considering the Hartree-Slater ionization background[22] (Fig. 2e). Further details of the fitting procedure are provided in the Methods section. In the following analysis, only the $L_3$ edge will be discussed due to the larger cross section (better spectral signal) compared the $L_2$ edge. Nevertheless, the mechanism which causes the splitting for both edges should be very similar[20]. The Gaussian fits of the Ti $L_3$ edges from various labeled positions displayed in Fig. 2e clearly show differences depending on the location within a vortex. On the one hand, at the vortex edge in the center of the PTO layer (where the polarization is large and points along the $c$ axis or [001]), the overall $L_3$ $e_g$ peak is asymmetric due to a more pronounced intensity of the $e_g'$ peak. On the other hand, at the vortex core, where the polarization along the $c$ axis is reduced, the overall $L_3$ $e_g$ peak is symmetric due to almost equal contributions coming from $e_g'$ and $e_g''$. Finally, within the essentially unpolarized STO layer there is a large contribution of $e_g'$ and a very small contribution from $e_g''$.

To investigate the spatial variation in the Ti $L$ edge, we created an EELS composition map using the Ti $L_{2,3}$ edge as shown in Fig. 3a, which provides atomic scale information about the location of Ti atomic columns in the PTO/STO superlattice. However, it fails to recognize the subtle variations in the Ti $L$ edge peak position within a vortex. To gain further insights, we summed STEM scan positions (binned) to the unit cell size (~6.4 Å) and fitted the STEM-EELS data with Gaussians after considering the Hartree-Slater background as shown in Fig. 2e and Supplementary Fig. 4.

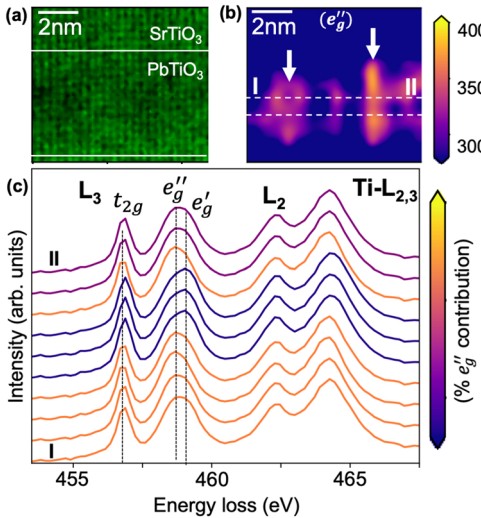

**Fig. 3 Atomic scale EELS mapping of polar vortices. a** Atomic resolution Ti L edge STEM-EELS map acquired from the same region as Fig. 2a. The locations of two vortices are indicated by dashed white circles. **b** Intensity of the $e_g''$ peak of the $L_3$ edge obtained by Gaussian fitting after binning by position. Increased intensity of the $e_g''$ is observed in the PTO vortex core, marked with white arrows. **c** EEL spectra extracted from along the I-II path indicated in **b**. At the vortex core, the contribution of $e_g''$ is the largest followed by the intermediate location and the vortex edge. A color bar guide indicating the relative amount of $e_g''$ contribution for **c** is displayed on the side.

The major differences of the EEL spectra are in the $L_3$ edge. We observe that the $e_g''$ contribution is highest at the vortex core (white arrows in Fig. 3b). The distance between the adjacent vortex cores is nearly 5 nm which agrees with the previously reported values[4] and our own data from Fig. 1. At the vortex edge (dark blue region in Fig. 3b between the white arrows), the $e_g''$ contribution is smaller. Due to relatively subtle differences between a vortex core and edge, the $e_g''$ intensity maps without binning (0.8 Å spatial resolution) have sparse high intensity signals. However, as we decrease the spatial resolution by binning to the size of the unit cell (~6.4 Å), the vortex core regions are better resolved (Supplementary Fig. 5). As shown in Supplementary Fig. 6, we do not lose any spectral information by decreasing the spatial resolution which implies that the hybridization changes are occurring at the unit cell length scale. We have performed a further level of discretization, extracting the spectra along the line from point I to II (Fig. 3c). The spectra are assigned a color according to the $e_g''$ contribution. The dominance of the $e_g''$ peak in the vortex core agrees with the results discussed in Fig. 2. Pardo et al.[28] measured line scans from similar vortex structures which match our results when we reduce the dimensionality of our data to 1D (see Supplementary Fig. 7). Our 2D mapping experiment with high spatial and energy resolution allows for the observation of subtle changes in the shape and position of the $t_{2g}$ and $e_g$ peaks in the presence of polar vortices.

**Interpretation of Ti $L$ edge fine structure**. A deeper understanding of how the strain and local polarization affect the electronic structure can be obtained from first-principles simulations. We carried out density-functional theory (DFT) calculations based on the local-density approximation (LDA) to ascertain the geometry and the electronic band structure of $(PbTiO_3)_6/(SrTiO_3)_6$ superlattices. The 6/6 calculations can be accurately extrapolated to interpret the experiments with the 16/16

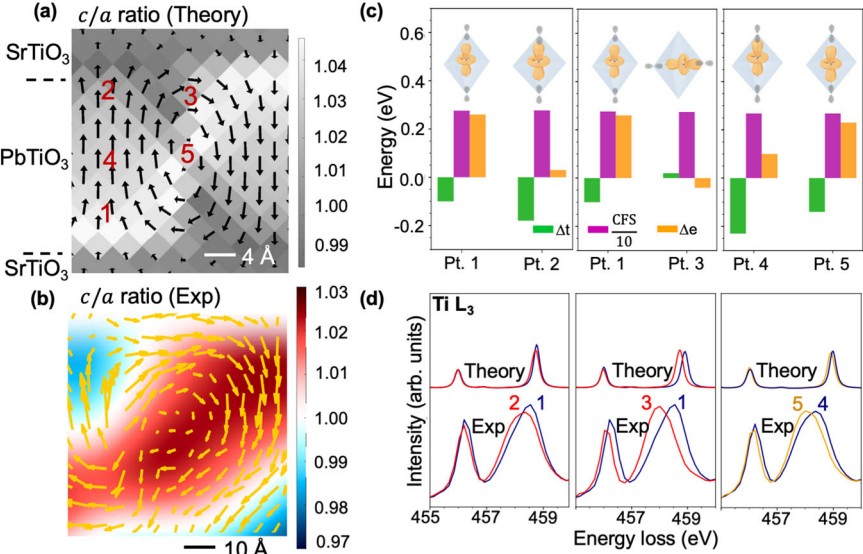

**Fig. 4 ab initio and multiplet calculations compared to experimental EELS. a** Theoretical local polarization profile of polydomain structures in PTO/STO superlattices. Background gray scale represents the local tetragonality. **b** Experimental polar displacement of A sites with c/a ratio maps overlaid in the background. **c** Change in crystal field parameters ($\triangle e_g$ in orange, $\triangle t_{2g}$ in green, and CFS in magenta) at five different positions of the $Ti^{4+}$ ions within the superlattice, as indicated in **a**. The inset shows the overlapping of the Ti $3d_{z^2}$ orbital and the O $2p$ orbital as the c/a ratio (points 1 and 2), the direction (points 1 and 3), and magnitude (points 4 and 5) of the $Ti^{4+}$ displacement. **d** Comparison of multiplet calculated EEL spectra using parameters in **b** as inputs and experimental EEL spectra for the five $Ti^{4+}$ positions.

periodicity, because from the structural point of view the structure we are analyzing in this work from first principles in 6/6 superlattices is essentially the same as the one obtained from second-principles simulations in 14/14 superlattices. (Supplementary Fig. 8) A full description of the methodology can be found in ref. [34]. From the structural point of view, the local polarization of the most stable polydomain configuration (Fig. 4a) displays alternating pairs of clockwise and counter-clockwise vortices along [100] in good agreement with the experimental images shown in Figs. 1, 4b and ref. [4,14]. The formation of vortices is associated with complex distortions of the lattice, and, in turn, these influence the local orbitals. From the eigenvalues and the electronic eigenfunctions we can project over specific atomic orbitals of specified quantum numbers to get a spatially resolved projected density of states (PDOS) localized around a region of interest[35] (see Supplementary Fig. 9). This local PDOS has been suggested to be an adequate quantity to compare ab initio simulations with EEL spectra[36].

The most important ingredients to understand the evolution of the experimental EEL spectra are: (i) the crystal-field splitting (CFS), defined as the difference between the average positions of the $e_g$ and $t_{2g}$ energy levels (see methods section); (ii) the $e_g$ splitting, $\triangle e_g = E(d_{x^2-y^2}) - E(d_{z^2})$; and (iii) the $t_{2g}$ splitting, $\triangle t_{2g} = E(d_{xy}) - \frac{E(d_{yz}) + E(d_{xz})}{2}$. The energy values associated with the different orbitals are computed from the centroid of the corresponding PDOS (see Supplementary Note 1). $\triangle e_g$ and $\triangle t_{2g}$ are strongly dependent on Ti $3d$-O $2p$ hybridization, which are very sensitive to the Ti-O bond length (details in Supplementary Fig. 10). We consider several factors influencing $\triangle e_g$ and $\triangle t_{2g}$. First, we study the effect of the change in tetragonality (the c/a ratio, see Supplementary Fig. 11) by comparing the $Ti^{4+}$ ions located at points 1 (c/a ≈ 1.03) and 2 (c/a ≈ 1.00) in Fig. 4a, b. The change in tetragonality at points 1 and 2 is related with the change in the local in-plane lattice constant, that is expected to be enlarged at the top interface (with respect to polarization direction) and compressed at the bottom interface. More details

are in the ref. [34]. We observe a significant reduction in $\triangle e_g$ (Fig. 4c) consistent with the asymmetric changes in equatorial and axial Ti-O overlap. Second, we observe the effect of the magnitude of the polarization by comparing what happens deep inside the polarization up domain at point 4 and the vortex core at point 5. We see that an increase in the polarization (see Supplementary Fig. 12) leads to an increase of the energy of antibonding axial orbitals ($3z^2 - r^2, xz, yz$) adding a negative contribution to both $\triangle e_g$ and $\triangle t_{2g}$. Third, we study the effect of polarization rotation by comparing points 1 and 3. If we consider bulk $PbTiO_3$, the electronic structure won't be affected by the change in the direction of Ti polarization. However, in the present case, due to the in-plane lattice constraints imposed by the DSO substrate, the local symmetry changes from tetragonal (out-of-plane Ti polarization) to orthorhombic (in-plane Ti polarization). More details in Supplementary Note 2 While the unit cell at location 1 is almost tetragonal, the unit cell at location 3 is orthorhombic expanding the unit cell in the equatorial plane with respect to the polarization. This lowers the energy of the equatorial orbitals (Supplementary Fig. 13) and yields a reduction of $\triangle e_g$. We note that the c/a ratio and the $Ti^{4+}$ polarization magnitude at points 1 and 3 are similar.

To understand how these changes are reflected in the EELS measurements, we have calculated the multiplet EEL spectra of the Ti L edge using our $\triangle e_g$, $\triangle t_{2g}$, and CFS as inputs. These calculated spectra were further compared with the experimental spectra extracted from equivalent points in Fig. 3. The small differences found in Fig. 4d between the two peaks in the Ti $L_3$ edges may be due to differences in the c/a ratio in the experimental and first-principles calculations (Fig. 4a, b).

## Discussion

In summary, monochromated STEM-EELS mapping at atomic resolution reveals the evolution of the Ti $3d$ orbital interactions as a function of their position in a vortex. Using this approach, we can detect subtle changes (~0.3 eV) in the $e_g$ and $t_{2g}$ peaks in Ti L

edge spectra within a vortex. As the polarization vector rotates, the axial Ti-O overlap changes due to a combination of the $c/a$ ratio and the magnitude and direction of the local dipoles around the $Ti^{4+}$ ions which influences the local $\triangle e_g$ and $\triangle t_{2g}$ splitting. The theoretically calculated crystal field multiplet EEL spectra agree with experimental EEL spectra and help us to understand this effect at a fundamental level. Finally, mapping of the Ti $3d$ orbitals will serve as a stepping-stone to understand the microscopic consequences of physical phenomena such as chirality and negative permittivity that have been reported in such polar textures.

## Methods

**Synthesis.** $[(PbTiO_3)_{16}/(SrTiO_3)_{16}]_8$ superlattices were synthesized on single-crystalline $DyScO_3$ $[110]_o$ substrates via reflection high-energy electron diffraction (RHEED)-assisted pulsed-laser deposition (KrF laser). The $PbTiO_3$ (PTO) and the $SrTiO_3$ (STO) layers were grown at 610 °C in 100 mTorr oxygen pressure. For all materials, the laser fluence was 1.0 J/cm² with a repetition rate of 10 Hz. RHEED was used during the deposition to ensure a layer-by-layer growth mode for both the $PbTiO_3$ and $SrTiO_3$. After deposition, the heterostructures were annealed for 10 min in 50 Torr oxygen pressure to promote full oxidation and then cooled down to room temperature at that oxygen pressure.

**STEM-EELS sample preparation and experimental conditions.** Cross-sectional samples of PTO/STO superlattices were mechanically polished using a 0.5° wedge. The samples were subsequently Ar ion milled in a Gatan Precision Ion Milling System, starting from 3.5 keV at 4° down to 1 keV at 1° for the final polish. The HAADF-STEM images were acquired using double aberration corrected TEAM I microscope operated at 300 kV under non-monochromated mode using a high angle annular dark field (HAADF) detector with a semi-convergence angle of 20 mrad and beam current of 70 pA. The LAADF-STEM images were acquired at a semi convergence angle of 10 mrad. For EELS mapping, we used monochromated STEM-EELS with an energy resolution of ~0.2 eV (measured by the full width at half maximum of the zero-loss peak) with a Gatan K3 camera installed in a Gatan Continuum GIF. The K3 camera is an electron counting detector that allows EELS measurement with high sensitivity and low noise. EEL spectrum images were collected with a 30 mrad STEM semi-convergence angle and 100 mrad EELS collection angle. The map was collected using 104 × 348 probe positions with a 0.8 Å step size and with a dwell time of 5 ms.

**Experimental STEM data processing**

*HAADF-STEM images.* The HAADF-STEM image from the region indicated in Fig. 1a was analyzed using custom Matlab scripts. Our imaging conditions were setup such that the perovskite A sites were visible and were used to analyze the A-site displacements as a function of location in the polar vortex. The STEM images were first drift corrected using the procedure described in ref. [27]. Then, each of the A sites was defined on a square lattice. Next, the subpixel position of each A site was fitted with a 2D Gaussian distribution where the local intensity $I$ over the coordinates $(x, y)$ for each peak was fitted using nonlinear least squares, given by the expression:

$$I(x, y) = I_0 \exp\left[-\frac{(x - x_0)^2 + (y - y_0)^2}{2\sigma^2}\right] + I_{BG},$$

where $I_0$ is the peak intensity, $\sigma$ is the peak standard deviation, $I_{BG}$ is the background offset, and $(x_0, y_0)$ are the peak's center coordinates. Next, we fitted the best-fit lattice vectors to all peaks, and then we calculated the displacement vectors $[u(x, y), v(x, y)]$ for all peaks from this best-fit lattice. These displacement vectors were then high-pass and low-pass filtered by a Gaussian filter with standard deviations of 1- and 4-unit cell lengths respectively, to best describe the local A site displacements. The local displacement vectors overlaid in Fig. 1c are scaled by 16x the true displacement. Finally, these displacement vectors were interpolated into a Cartesian grid with unit cell spacing, and then differentiated to obtain the infinitesimal strain tensor. The component of the strain tensor that is most correlated with the vortex structure is the infinitesimal rotation $\theta$, which is given by the expression

$$\theta = \frac{1}{2}\left(\frac{\partial u}{\partial y} - \frac{\partial v}{\partial x}\right).$$

This infinitesimal rotation is plotted in Fig. 1c, where the color channels indicate the magnitude of local A-site infinitesimal rotation, and the intensity is scaled by the local mean peak intensity. This intensity mask removes the signals from the lower intensity STO layers, showing the vortex domains inside the PTO layers clearly. Our method is similar to the widely reported geometric phase analysis (GPA)[37,38].

*EELS mapping and spectra.* The spectra were denoised using the non-negative matrix factorization (NMF) algorithm in Hyperspy based on a Poisson noise

spectrum[38]. The first 3 components of the NMF output were summed to get the de-noised spectra. To fit the peak, the EELS map was binned to different pixel sizes ($0 \times 0$, $2 \times 2$, $4 \times 4$, $8 \times 8$). The Ti L edge at each position in the EELS map was fitted using a model fitting algorithm. The steps were as follows:

1. The Hartree-Slater ionization edges were identified from the average spectrum. The ionization edge typically corresponds to 10% of the maximum intensity in the $L_3/L_2$ edge.
2. The $L_3/L_2$ ratio was fixed to 2:1 to account for the scattering cross-sections.
3. After the Hartree-Slater background was fixed for all probe positions, Gaussian fits were performed by imposing bounds on the center of the $L_3$ and $L_2$ peak.

**Crystal field multiplet calculations**

*$e_{g}$, and $t_{2g}$ splitting values.* The spectral centroid was calculated by taking the weighted average for all the peaks present in the PDOS spectrum for all $d$-orbitals (see Supplementary Fig. 9). The integrals required to compute the center of mass of the different PDOS were obtained using the "scipy" integration algorithm[39]. Once, a singular energy value was found for all the $d$ orbitals, the $e_g$ ($E(e_g)$) and $t_{2g}$ ($E(t_{2g})$) splitting values were calculated using the equations below:

$$\Delta e_g = E(d_{x2-y2}) - E(d_{z2})$$

$$\Delta t_{2g} = E(d_{xy}) - \frac{(E(d_{yz}) + E(d_{xz}))}{2}$$

where $E(d)$ is the spectral centroid calculated from the density of states of the $d$ orbital.

The crystal field splitting was then calculated by the equation:

$$CFS = 10D_q = E(e_{g,avg}) - E(t_{2g,avg})$$

where $E(e_{g,avg})$ and $E(t_{2g,avg})$ are defined as:

$$E(e_{g,avg}) = (E(d_{x2-y2}) + E(d_{z2}))/2$$

$$E(t_{2g,avg}) = (E(d_{xy}) + E(d_{xz}) + E(d_{yz}))/3$$

*Crystal field multiplet calculations.* Crystal field multiplet calculations were performed using CTM4XAS software by Degroot[40]. The software requires input parameters of $D_s$, $D_t$ and Dq. 'Dq' was calculated from the crystal field splitting. For the $D_s$ and $D_t$ values the following equations were adopted,

$$D_s = \frac{E(e_g) + E(t_{2g})}{7},$$

$$D_t = \frac{3D_s - E(t_{2g})}{5}.$$

## Data availability

Any additional data required to evaluate the paper can be requested from corresponding authors.

## Code availability

The MATLAB-based toolbox for fitting atom positions and calculating local polarization is available from corresponding authors upon request. Similarly, the python-based scripts for EELS spectrum analysis and multiplet calculations can be requested from corresponding authors.

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

## Acknowledgements

The experiments were performed at the Molecular Foundry, Lawrence Berkeley National Laboratory, which is supported by the U.S. Department of Energy under contract no. DE-AC02-05CH11231. S.S. is supported by the DOE Quantum Materials project. JJ and PGF acknowledge financial support from Grant PGC2018-096955-B-C41 funded by MCIN/AEI/ 10.13039/501100011033. P. A.-P. acknowledges support from the Diputación de Gipuzkoa under grant 2020-FELL-000005-01. L.W.M. acknowledges support from the National Science Foundation under Grant DMR-1708615. C.O acknowledges support from the DOE Early Career Research Award program. S.S. thanks J. Ciston and C. Song for the technical assistance during EELS measurements. S.S. also thanks M. McCarter for useful scientific discussions.

## Author contributions

S.S. performed the HAADF-STEM, EELS experiments and EELS multiplet calculations. P.G.F. performed the first principles PDOS calculations on pure PbTiO$_3$ and vortices under different conditions of strain and polarization. C.O. performed the polarization analysis of the STEM images. S.D. grew the samples by PLD. P.A.P. contributed to first principles discussion. M.M. provided the XAS experimental data. P.E. supervised the EELS experiments and helped in the experimental EELS data analysis. L.W.M. participated in the technical discussion. R.R. and J.J. envisioned the idea and supervised the project. All authors contributed equally to the writing of the manuscript.

## Competing interests

The authors declare no competing interests.
