## [Peer Review File · Nature Communications]

REVIEWER COMMENTS

Reviewer #1 (Remarks to the Author):

Referee Reports:

Ref. NCOMMS-21-20807

In this work, the authors report an atomic level study on the crystal field mapping in the polar vortex phase in the PTO/STO 16 by 16 superlattice grown on a DSO substrate. The spectrometer on the basis of direct electron detector was used for obtaining the EELS signals, which is better for further justification of the EELS signals. The recorded EELS signals here indicate subtle electronic structure changes inside the vortex phase. I raise the following concerns that the authors should address before the paper is ready for further consideration.

1. The author state that 'the magnitude and direction of the local dipoles around the Ti^{4+} ions which influences the local Δ_{eg} and Δ_{t2g} splitting'. Such a description is rather confusing since the directions of dipoles might not change the electronic structure of a PTO unit cell. I cannot agree with the authors that the EELS characterization has potential to detect only the polar axis of a PTO unit cell. I think that the EELS spectra should be the same for two PTO unit cells which have in-plane and out-of-plane polarizations with the same magnitude.

2. Do the unit cells around a vortex core show the same EELS signals? Is the polarization magnitude for the in-plane polar area around a core the same as that of the out-of-plane? How about their EELS signals?

3. The schematic representing the TiO_6 in Fig. 1(d) is confusing for the upper right panel TiO_6 , which rotates 45° relative to its matrix PTO unit cell. This is almost impossible here since the PTO and STO matrix exhibit no O octahedral rotation.

4. In Figure S1, it is difficult to understand the mapping of E_{xx} . For a PTO layer with the vortex pattern, the regions near the PTO/STO interfaces possess horizontal polarizations and E_{xx} should be tensile. The region in the interior of the PTO layer possesses vertical polarizations and E_{xx} should be compressive. The mapping of E_{xx} is just opposite. The authors should double-check their results. I am not sure what kind of strain mapping method was used in Fig. S1, but, it seems that the authors have confused the in-plane and out-of-plane strain maps. Here the GPA (geometric phase analysis) might be suitable for extracting such strains with high precision, as successfully used in the studies of polar topologies such as Ref. 3 in the manuscript; Nature Communications 8:15994 (2017); and Nature Materials 19, 881–886 (2020).

5. It is not clear whether it is [011] $DyScO_3$ substrate was used. I note the [011] direction is actually corresponding to some very unusual directions, which is impossible for preparation of such [001]pc oriented PTO/STO superlattice.

6. The author declare that a -0.43% compressive strain was exerted on PTO unit cells. I think this calculation method for the strain is not appropriate for this case and will cause potential confusing in the study of polar topologies in ferroelectrics. Actually here the tensile strain was exerted for PTO layer, so that the vortex phase containing larger amount of in-plane polarization domains forms. If compressive strains were exerted, the out-of-plane polarizations would be promoted. This is also clear in Ref. 3 since tensile strains promote the larger amount of in-plane polarization domains in the flux-closure phase. If the PTO layers are thick enough, periodically distributed a/c domains may form in these multilayer structures (Ref. 3), which is a strong signature that the PTO layers are tensile strained.

7. The $(PbTiO_3)_6/(SrTiO_3)_6$ superlattice, used in the first-principles calculations, remarkably differs from the experimental 16 by 16 superlattice. I suggest the authors should rationalize it.

8. In Figure 4(a) on Page 12, there is an asymmetry in the c/a ratio mapping: The point 1 (polarization from STO to PTO) has a larger c/a ratio than the point 2 (polarization from PTO to STO). Could the author explain why this happens?

X.L.Ma

Reviewer #2 (Remarks to the Author):

The authors have investigated the electronic structure of polar vortices in $[(\text{PbTiO}_3)_{16}/(\text{SrTiO}_3)_{16}]$ superlattices at the atomic scale by using a combination of energy loss near-edge structure analysis, crystal field multiplet theory, and first-principles calculations. It is found that the peaks in Ti L-edge EEL spectra shift systematically depending on the direction and magnitude of the local dipole. First-principles computation of the local projected density of states on the Ti 3d orbitals and the simulated crystal field multiplet spectra derived from first-principles are in good agreement with the experiments. The results are somewhat help for understanding the electronic structure and physical properties of polar vortices. However, the work is not significant enough to be published in the high impact journal like Nature Communications. My detailed comments are as follows:

1.The authors have employed the monochromated STEM-EELS mapping at atomic resolution to reveal the evolution of the Ti 3d orbital interactions as a function of their position in a vortex. It is obvious that the direction and magnitude of the local dipole in a polar vortex can change the orbital interactions. □

2.The authors claim that mapping of the Ti3 d orbitals will serve as a stepping stone to understand the microscopic origins of physical phenomena such as chirality and negative permittivity of polar textures. The formation of chiral vortex is usually due to the long-range electrostatic and elastic interactions between different dipoles in the superlattices, in which the change of orbital interactions is not necessary. What's the relationship between the orbital interactions and the chirality of the vortex?

3.The detailed information of the first-principles calculations is absent in the manuscript. The periodic boundary conditions are usually employed in first-principles calculations. In the model of Fig.4(a), the boundary conditions are not periodic. How do the authors obtain the vortex of Fig.4(a) from first-principles calculations?

4.In the first-principles calculations of Fig.S9, do the authors consider other distortion like oxygen octahedra rotation?

Reviewer #3 (Remarks to the Author):

The manuscript presents monochromated STEM-EELS mapping at atomic resolution to reveal the evolution of the Ti 3d orbital interactions as a function of their position in a vortex. The theoretically calculated crystal field multiplet EEL spectra are in agreement with experimental EEL spectra and help us to understand the effect at a fundamental level. The research was done in a comprehensive way, and might be suitable for publication after some revision with the following issues.

Deeper understanding of how the strain and local polarization affect the electronic structure could be obtained from first-principles simulations. The authors carried out density-functional theory (DFT) calculations based on the local-density approximation (LDA) by building $(\text{PbTiO}_3)_6/(\text{SrTiO}_3)_6$ superlattices. The in-plane lattice parameters and unit cell numbers should be provided. From Figure S8, the supercell used could not contain any polar vortex.

In Figure 4, no scale bars are given for (b), so it is unclear that whether (a) and (b) have same length scale.

Note that the authors indicated that one vortex has a dimension of about 10nm in the Introduction. Also note that $[(\text{PbTiO}_3)_{16}/(\text{SrTiO}_3)_{16}]$ superlattices (rather than $(\text{PbTiO}_3)_6/(\text{SrTiO}_3)_6$) are synthesized in experiment.

Upon closer inspection, (a) and (b) show significant difference on the forms of the vortices, see the pattern of arrows and (gray) color codes of (a) and (b). The vortex in (a) seems to be more symmetric than (b), any explanation on this?

The authors constructed the corresponding A site displacement vector maps to indicate the location of the vortex cores etc. How about the B site displacement for such vortex determination?

The authors made careful comparison of Ti^{3+} and Ti^{4+} reference spectra, indicating absence of the possibility of Ti^{3+} in the system. As both Pb and Sr are bivalent elements, what is the reason to consider the Ti^{3+} state?

Minor points,

STO adopts a $Pm\bar{3}m$ cubic space group, the symbol of bar should be on top of 3, instead of an underline?

Line 122, ... Ti L edge spectra at the vortex core as compared to Ti^{4+} (ref) indicates a negligible possibility of the presence of Ti^{3+} in our samples. What is the meaning of ref here?

REVIEWER COMMENTS

Reviewer #1 (Remarks to the Author):

Referee Reports:

Ref. NCOMMS-21-20807

X.L.Ma

In this work, the authors report an atomic level study on the crystal field mapping in the polar vortex phase in the PTO/STO 16 by 16 superlattice grown on a DSO substrate. The spectrometer on the basis of direct electron detector was used for obtaining the EELS signals, which is better for further justification of the EELS signals. The recorded EELS signals here indicate subtle electronic structure changes inside the vortex phase. I raise the following concerns that the authors should address before the paper is ready for further consideration.

Response:

We would like to thank the reviewer for taking time to read the manuscript and appreciating our work. We have carefully answered all the concerns that the reviewer was pointing to.

Question 1.1 The author state that ‘the magnitude and direction of the local dipoles around the Ti⁴⁺ ions which influences the local Δ_{eg} and Δ_{t2g} splitting’. Such a description is rather confusing since the directions of dipoles might not change the electronic structure of a PTO unit cell. I cannot agree with the authors that the EELS characterization has potential to detect only the polar axis of a PTO unit cell. I think that the EELS spectra should be the same for two PTO unit cells which have in-plane and out-of-plane polarizations with the same magnitude.

Response:

We thank the reviewer for bringing up one of the most important points in the manuscript. Indeed, this aspect was already discussed in the supplementary material, **Fig. S13**.

In summary, we agree with the reviewer that if we consider bulk PbTiO₃, the electronic structure won't be affected by the change in the direction of Ti polarization. However, in the present case, due to the in-plane lattice constraints imposed by the DSO substrate, the local symmetry changes from tetragonal (out-of-plane Ti polarization) to orthorhombic (in-plane Ti polarization). This change in the local orthorhombicity affects the corresponding electronic structure. We detect these subtle electronic structure changes from EELS. Details copied from the supplementary information are described below:

Changes in the manuscript:

Supplementary Information

“In PTO/STO superlattices a third effect becomes important which is the rotation of the polarization. Performing the same kind of analysis as above is not possible as it is important to project the density of states on orbitals with the correct quantization axis. Thus, in the previous tetragonal situations the z axis was the main symmetry axis and projection was carried out on the $3z^2-r^2$, x^2-y^2 , xy , xz and yz orbitals. However, if the polarization is observed above or below the vortex core (see **Fig. 4** of the main body of the manuscript) we can see that the polarization points in the x direction and the correct orbitals to project the density of states would be $3x^2-r^2$, z^2-y^2 , zy , xz and yz . In order to avoid problems with the quantization axis we analyze the initial situation (polarization pointing in the direction of the positive z axis) and the final one (polarization pointing in the direction of the positive x axis) without discussing intermediate geometries. The first situation (polarization along z) can be simply represented by the previous calculations when $\lambda=1$.

Moreover, in bulk, when the polarization points along the x -axis, the situation is completely equivalent to the case when it points along z . However, in DSO-grown PTO/STO superlattices the in-plane lattice constants (a, b) are fixed by the substrate. Thus, the lattice constant in x and y is fixed to that of DSO while the one along z can be accommodated to minimize the energy of the system. That means that when the polarization runs along z the equatorial (with respect to the polarization) lattice constants are fixed while the longitudinal one is free to change, while in the case when it runs along x the lattice constants that are fixed are the longitudinal and one equatorial, while the second equatorial is free to change (see **Fig. S13(a)**). If we now take the $\lambda=1$ case as a reference and we increase an equatorial lattice constant to simulate what is happening above the vortex core we can see that the x^2-y^2 orbital energy decays sharply leading to a negative contribution to Δe_g as shown in **Fig. S13(b)**.

Figure S13. (a) Illustration of the initial and final geometries as the polarization rotates from the z to the x axis. In **red** we show the constraints imposed by the substrate to the lattice constant the direction along which the lattice constant is free to change is shown in **green**. As the system rotates the polarization from the z -axis to the x -axis it goes

from tetragonal (z-axis) to orthorhombic (x-axis) due to the constraints imposed on the lattice constants by the substrate. (b) Variation of the band center-of-mass as the orthorhombicity of the system is changed (the lattice constant b is increased keeping a and c fixed).

Main Text

...ng points 1 and 3. If we consider bulk PbTiO_3 , the electronic structure won't be affected by the change in the direction of Ti polarization. However, in the present case, due to the in-plane lattice constraints imposed by the DSO substrate, the local symmetry changes from tetragonal (out-of-plane Ti polarization) to orthorhombic (in-plane Ti polarization). While the unit cell....

Question 1.2 Do the unit cells around a vortex core show the same EELS signals? Is the polarization magnitude for the in-plane polar area around a core the same as that of the out-of-plane? How about their EELS signals?

Response:

No, the region around the vortex core does not show the same EELS signals. There are some points where the polarization magnitude around the vortex core for in-plane and out-of-plane is the same, but the EELS signals are different. We have made this comparison in Figure 4c (middle panel; marked as red block). From both theory and experiments, the EELS spectra are different for regions with the same magnitude of in-plane and out-of-plane polarization.

Changes in the manuscript:

Main text:

Figure 4 Ab-initio and multiplet calculations (c) Change in crystal field parameters (Δe_g in orange, Δt_{2g} in green, and CFS in magenta) at five different positions of the Ti^{4+} ions within the superlattice, as indicated in (a). The inset shows the overlapping of the Ti $3d_{z^2}$ orbital and the O $2p$ orbital as the c/a ratio (points 1 and 2), the direction (points 1 and 3), and magnitude (points 4 and 5) of the Ti^{4+} displacement. (d) Comparison of multiplet calculated EEL spectra using parameters in (b) as inputs and experimental EEL spectra for the five Ti^{4+} positions.

Question 1.3 The schematic representing the TiO_6 in Fig. 1(d) is confusing for the upper right panel TiO_6 , which rotates 45° relative to its matrix PTO unit cell. This is almost impossible here since the PTO and STO matrix exhibit no O octahedral rotation.

Response:

We apologize for the confusing schematic in the previous version of the manuscript and thank the referee for pointing this out. The reviewer correctly indicates that the octahedron on the top-right corner of the panel displayed an unphysical rotation that did not correspond to the actual distortion at the vortex. In the new version of the schematic in Fig. 1(d) we corrected this error by using actual coordinates extracted from the first-principles simulations. In this new version all octahedra stay essentially upright and it is the displacement vector of the Ti atom inside the O cage which rotates. Here is the new schematic.

Changes in the manuscript:

Main text:

Question 1.4 In Figure S1, it is difficult to understand the mapping of E_{xx} . For a PTO layer with the vortex pattern, the regions near the PTO/STO interfaces possess horizontal polarizations and E_{xx} should be tensile. The region in the interior of the PTO layer possesses vertical polarizations and E_{xx} should be compressive. The mapping of E_{xx} is just opposite. The authors should double-check their results. I am not sure what kind of strain mapping method was used in Fig. S1, but, it seems that the authors have confused the in-plane and out-of-plane strain maps. Here the GPA (geometric phase analysis) might be suitable for extracting such strains with high precision, as successfully used in the studies of polar topologies such as Ref. 3 in the manuscript; Nature Communications 8:15994 (2017); and Nature Materials 19, 881–886 (2020).

Response:

We want to thank the reviewer for pointing this out this mistake. Figure S1 caption has the in-plane and out-of-plane labels reversed. We have fixed this unfortunate typo.

Changes in the manuscript:

Figure S1: HAADF STEM image of PTO/STO superlattices with the corresponding out of plane and in-plane strain maps respectively

We first drift corrected the image using method described in ref 27 of the manuscript. We fit each of the A site positions using a 2D Gaussian function. Next, these sites were indexed and the displacements relative to the mean lattice coordinates were calculated. Using a smooth interpolation method (kernel density estimation), these displacement values at each A site position were used to calculate displacement images at the same resolution as the original image. Finally, the strain tensor components were computed from numerical differentiation of these 2D displacement images (in the same manner as GPA). This procedure is outlined in Part 3 of the supplementary section of our manuscript. In this section, we have now specifically mentioned that our method is similar to geometric phase analysis and added the citations.

Changes in the manuscript:

...layers clearly. Our method is similar to the widely reported geometric phase analysis (GPA).
1,2
....

Question 1.5 It is not clear whether it is [011] DyScO₃ substrate was used. I note the [011] direction is actually corresponding to some very unusual directions, which is impossible for preparation of such [001]_{pc} oriented PTO/STO superlattice.

Response:

We apologize for this mistake in notation. We are referring to [110]_o DSO substrate We have corrected this in the new version.

Changes in the manuscript:

....DyScO₃ [110]_o substrates via reflection high-energy electron diffraction (RHEED)-assisted

Question 1.6 The author declare that a -0.43% compressive strain was exerted on PTO unit cells. I think this calculation method for the strain is not appropriate for this case and will cause potential confusing in the study of polar topologies in ferroelectrics. Actually here the tensile strain was exerted for PTO layer, so that the vortex phase containing larger amount of in-plane polarization domains forms. If compressive strains were exerted, the out-of-plane polarizations would be promoted. This is also clear in Ref. 3 since tensile strains promote the larger amount of in-plane polarization domains in the flux-closure phase. If the PTO layers are thick enough, periodically distributed a/c domains may form in these multilayer structures (Ref. 3), which is a strong signature that the PTO layers are tensile strained.

Response:

This depends on how the strain is defined. It is possible to define the misfit strain with respect to a different reference frame. In the present paper, we decided to use the cubic

lattice parameter extrapolated to room temperature. For this reference, PbTiO_3 is under compressive strain on DyScO_3 , for example.

Another choice would be to define the strain with respect to the 0 K lattice parameters of the $P4mm$ phase found by minimizing the Helmholtz free energy, which includes the zero-point energy corrections from vibrational degrees of freedom. With this criterion, the PbTiO_3 is under expansive tensile strain. This is the criterion chosen by E. Ritz and N. Benedek, *Phys. Rev. Mater.* **4**, 084410 (2020) in their paper “Strain game revisited for complex oxide thin films: Substrate-film thermal expansion mismatch in PbTiO_3 ”. In the revised version of the manuscript, we have changed the reference, according to the request of the reviewer.

Changes in the manuscript:

First, the epitaxial strain imposed by the DSO substrate, ($a_{pc} = 3.952 \text{ \AA}$ where pc stands for pseudo-cubic) translates into tensile strains of +1.20 % on STO, and +1.3 % on PTO, where the epitaxial strain has been computed with respect the cubic lattice parameters of SrTiO_3 and the tetragonal phase of PbTiO_3 at room temperature.

Question 1.7 The $(\text{PbTiO}_3)_6/(\text{SrTiO}_3)_6$ superlattice, used in the first-principles calculations, remarkably differs from the experimental 16 by 16 superlattice. I suggest the authors should rationalize it.

Response:

We thank the reviewer for this interesting question.

It is true that the first-principles calculations were carried out in a $(\text{PbTiO}_3)_6/(\text{SrTiO}_3)_6$ superlattice, mostly due to computational efficiency. The simulation box required to include the vortices in a 6x6 superlattice already amounts to 720 atoms resulting in a long computation time. Changing to a 16x16 superlattice would make the calculations prohibitively computationally expensive. Further, the 6x6 calculations can be accurately extrapolated to interpret the experiments with the 16/16 periodicity because from the structural point of view, the structure we are analyzing in this work from first-principles calculations in 6/6 superlattices is essentially the same as the one obtained from second-principles simulations in 14/14 superlattices, where we can complete relaxations of much larger supercells. As shown in the Fig. S8, the presence of ordered arrays of clockwise/counter-clockwise vortices within the PbTiO_3 layer is clearly reproduced in both cases. Unfortunately, the current version of our second-principles code does not allow the computation of the band structures, the basic ingredient to analyze the EELS spectra.

Figure S8 Simulations on the same $\text{PbTiO}_3/\text{SrTiO}_3$ superlattice system using different methods. Left: first-principles simulations in a 6/6 superlattice. Right: second-principles method using a 14/14 superlattice.

Therefore, to deal with the electronic structure we have to resort to the first-principles simulations in a 6/6 superlattice. However, our experience in similar superlattices shows how the layer-by-layer electronic structure rapidly converges to a constant bulk-like value (under the same boundary conditions) when we move away from the superlattice. We show in Fig. R1, for instance, the layer-by-layer projected density of states (PDOS) of a $\text{PbTiO}_3/\text{SrTiO}_3$ capacitor. Just two unit cells away from the interface (bottom panel in Fig. R1) the PDOS has converged to the bulk values. In the case of insulator/insulator superlattices, the situation is even clearer. There, the normal component of the dielectric displacement has to be preserved. This continuity of PDOS makes the force-constant matrix of the quasi-one-dimensional super-lattice short-ranged in real space. This “locality principle” implies that one may expect the individual electronic properties and layer polarizations to depend only on the local compositional environment comprising the layer itself and few nearby neighbors.

As a consequence, the inclusion of more unit cells would produce a projected density of states that are essentially the same one unit cell after the other.

[Redacted]

Figure R1 PDOS of the inequivalent TiO₂ layers in the unpolarized PbTiO₃/SrTiO₃ capacitor (solid curves with gray shading). The bottom curve lies next to the electrode, and the top one lies in the center of the PbTiO₃ film. Only the PDOS on half of the symmetric supercell are shown. The bulk PDOS curves (red dashed) are aligned to match the Ti(3s) peak at $E \sim -57$ eV. The Fermi level is located at zero energy. Figure taken from M. Stengel, P. Aguado-Puente, N. Spaldin, and J. Junquera, Phys. Rev. B **85**, 184105 (2012).

Changes in the manuscript:

Main Text:

.....superlattices. The 6x6 calculations can be accurately extrapolated to interpret the experiments with the 16/16 periodicity because from the structural point of view, the structure we are analyzing in this work from first-principles in 6/6 superlattices is essentially the same as the one obtained from second-principles simulations in 14/14 superlattices. (**Fig. S8**) A full description of the methodology can be found in Ref. ³⁴. From the structural point of view, the lo.....

Supplementary Information:

First-principles calculations were carried out in a (PbTiO₃)₆/(SrTiO₃)₆ superlattice, mostly due to computational efficiency. The simulation box required to include the vortices in a 6/6 superlattice already amounts to 720 atoms resulting in a long computation time. Changing to a 16/16 superlattice would make the calculations prohibitively computationally expensive. Further, the 6/6 calculations can be accurately extrapolated to interpret the experiments with the 16/16 periodicity because from the structural point of view, the structure we are analyzing in this work from first-principles in 6/6 superlattices is essentially the same as the one obtained from second-principles simulations in 14/14 superlattices, where we can complete relaxations of much larger supercells. As shown in the **Fig. S8**, the presence of ordered arrays of clockwise/counter-clockwise vortices within the PbTiO₃ layer is clearly reproduced in both cases. Unfortunately, the current version of our second-principles code does not allow the computation of the band structures, the basic ingredient to analyze the EELS spectra.

Question 1.8 In Figure 4(a) on Page 12, there is an asymmetry in the c/a ratio mapping: The point 1 (polarization from STO to PTO) has a larger c/a ratio than the point 2 (polarization from PTO to STO). Could the author explain why this happens?

Response:

We thank the reviewer for this interesting observation. It is addressed in Section III C of the paper by P. Aguado-Puente and J. Junquera, Phys. Rev. B **85**, 184105 (2012). More precisely, in Fig. R2, R3, and R4. It is related with the change in the local in-plane lattice constant, that is expected to be enlarged at the top interface (with respect to polarization direction) and compressed at the bottom interface. This correlates with the tetragonality: the PbTiO₃ unit cells close to the bottom interface (with respect to the polarization direction) are compressed in-plane, and as a result, they tend to elongate

along the z axis. Conversely, at the top interface, the material is expanded in-plane and presents a reduced tetragonality.

We just reproduce here the most relevant paragraphs of the former reference:

“The analysis of the polarization and tetragonality profiles reveals that the formation of domains in the superlattices is associated with complex distortions. The characteristics of the strain field in this system can be explained as a combination of different effects.

On the one hand, in PbTiO_3 the off-center displacements of both the Pb and Ti cations contribute to the polarization. Therefore the Pb atoms displace along z in opposite direction in the up and down domains. This gives rise to an offset between $[100]$ atomic rows to the left and right of the DW [see Fig. R2(a)]. A sizable offset of 0.6 \AA was already predicted by Meyer and Vanderbilt in 180° stripe domains in bulk PbTiO_3 . As in Ref. [B. Meyer and D. Vanderbilt, Phys. Rev. B **65**, 104111 (2002)], for the $\text{PbTiO}_3/\text{SrTiO}_3$ superlattices we quantify this offset for a given layer as the difference in the z coordinate of a equivalent A cation at the center of opposite domains. The layer by layer offset, shown in Figs. R3(a) and R3(c), amounts up to almost 0.5 (0.45) Å at the middle of the PbTiO_3 layer in the $(6 \mid 6) [(3 \mid 3)]$ superlattice. Although the offset of opposite domains is partially accommodated by the interfaces—which reflects in the increase (decrease) of the tetragonality at the bottom (top) interface in Figs. R4(b) and R4(d)—it still propagates into the SrTiO_3 , amounting a sizable $\sim 0.1 \text{ Å}$.

[Redacted]

Figure R2 Schematic representation of the distortion induced by the domain structure in (a) bulk PbTiO_3 , (b) PbTiO_3 thin films and (c) $\text{PbTiO}_3/\text{SrTiO}_3$ superlattices. (a) In bulk, displacements of Pb cations cause an offset between $[100]$ atomic rows across the DW. (b) In thin films, in addition to the offset between domains, rotation of the polarization near the interface is responsible of a nonvanishing strain gradient. (c) In the case of the $\text{PbTiO}_3/\text{SrTiO}_3$ superlattices, the offset and modulation of the strain field in the PbTiO_3 layer (in grey) propagates into the SrTiO_3 (in white).

On the other hand, in thin films the polarization in PbTiO_3 rotates at the DW. Indeed, as pointed out above, our simulations show that large in-plane displacements of the Pb

atoms (up to 0.2 Å) take place at the interfaces. It is sensible to argue that this in-plane polarization is coupled with an in-plane strain and, as it is schematically depicted in Fig. R2(b), it pushes the DW in the same direction of the polarization. This effect is reinforced as consecutive DWs become closer, as it happens in ferroelectric thin films [see Fig. R2(b)]. As a consequence, **the in-plane lattice constant is expected to be enlarged at the top interface (with respect to polarization direction) and compressed at the bottom interface.** To test this hypothesis we have performed a detailed analysis of the strain field in the system, calculating for every individual perovskite unit cell the local values of the in-plane lattice constant, a . The local in-plane strain, calculated as $\varepsilon_{11} = a/a_0 - 1$, where $a_0 = a_{\text{SrTiO}_3} = 3.874 \text{ \AA}$ is plotted in Figs. R3(b) and R3(d) for the (3 | 3) and (6 | 6) superlattices, respectively. **It shows a variation with respect to the position along the z direction that can be clearly correlated with that of the tetragonality**, shown in Figs. R4(b) and R4(d): PbTiO₃ unit cells close to the bottom interface (with respect to the polarization direction) are compressed in-plane, and as a result, they tend to elongate along the z axis. Conversely, at the top interface, the material is expanded in-plane and presents a reduced tetragonality.

[Redacted]

Figure R3. Left panels: local layer-by-layer offset between [100] atomic rows to the left and right of the DW for (a) a (3 | 3) and (c) a (6 | 6) superlattice. Right panels: local in-plane strain across the center of an up domain in (b) a (3 | 3) and (d) a (6 | 6) superlattice. A large nondiagonal component of the strain gradient, can be observed close to the interfaces.

[Redacted]

Figure R4 Left panels: layer-by-layer out-of-plane polarization, P_z , inferred from the Born effective charges and the atomic displacements for (a) a (3 | 3) and (c) a (6 | 6) superlattice. Right panels: layer-by-layer tetragonality for (b) a (3 | 3) and (d) a (6 | 6) superlattice. Empty symbols represent values at the center of an up domain, while filled symbols correspond to averaged values (root mean square in the case of polarization) along the [100] direction.

For the reader's convenience, we have now included a sentence in the discussion of the main manuscript.

Changes in the manuscript:

Main Text:

by comparing the Ti^{4+} ions located at points 1 ($c/a \approx 1.03$) and 2 ($c/a \approx 1.00$) in Fig. 4(a) and Fig. 4(b). The change in tetragonality at point 1 and 2 is related with the change in the local in-plane lattice constant, that is expected to be enlarged at the top interface (with respect to polarization direction) and compressed at the bottom interface. More details are in the ref ³⁴. We observe a significant reduction in Δe_g (Fig. 4(c)) consistent with the asymmetric changes in equatorial and axial

X.L.Ma

Reviewer #2 (Remarks to the Author):

The authors have investigated the electronic structure of polar vortices in [(PbTiO₃)₁₆/(SrTiO₃)₁₆] superlattices at the atomic scale by using a combination of energy loss near-edge structure analysis, crystal field multiplet theory, and first-principles calculations. It is found that the peaks in Ti L-edge EEL spectra shift systematically depending on the direction and magnitude of the local dipole. First-principles computation of the local projected density of states on the Ti 3d orbitals and

the simulated crystal field multiplet spectra derived from first-principles are in good agreement with the experiments. The results are somewhat help for understanding the electronic structure and physical properties of polar vortices. However, the work is not significant enough to be published in the high impact journal like Nature Communications.

Response:

We would like to politely disagree with the reviewer. Although polar vortices and skyrmions are being well studied, so far, the microscopic details of the d -orbitals in the Ti^{+4} ions within the vortex have not been reported. In this work for the first time, atomic-resolution EELS, first-principles and the multiplet calculations were used together to understand the electronic structure within the vortex. Additionally, this is the first work that discusses the orbital rotation and successfully experimentally probes it with monochromated, atomic resolution EELS mapping. Orbital rotation is a fundamental effect that is required to explain the origins of chirality in this system. Lovesey and colleagues ([Phys. Rev. B **98**, 155410 (2018); Ref. [19] of the present manuscript) theoretically proposed that the presence of chirality is due to a change in the quadrupole moments of the Ti 3d orbitals. These quadrupole moments arise because of the orbital hybridization. Thus, the present work focuses on measuring the orbital hybridization at the atomic scale serves as a first step to validate this theory experimentally. Therefore, in summary, this paper is fundamentally important to reveal the atomic scale origins of the physical phenomena that have been reported for such vortices and skyrmions.

My detailed comments are as follows:

Question 2.1 The authors have employed the monochromated STEM-EELS mapping at atomic resolution to reveal the evolution of the Ti 3d orbital interactions as a function of their position in a vortex. It is obvious that the direction and magnitude of the local dipole in a polar vortex can change the orbital interactions. □

Response:

We thank the referee for making this point. While it is perhaps obvious, after the fact, that such a correlation should exist, to the best of our knowledge, this has not been reported before. For this reason, at the time of writing our manuscript, the answer to this question was not as obvious as assumed by the reviewer. Indeed, we would like to point him/her to the first question by the first reviewer of the present work, where he/she claims that “I think that the EELS spectra should be the same for two PTO unit cells which have in-plane and out-of-plane polarizations with the same magnitude.” The clarification of this point from a combination of state-of-the-art experimental and theoretical techniques, which allows the measurement of the EELS spectra with an unprecedented resolution, is the strongest point of our work.

The disentanglement of the different ingredients (polarization and its interplay with the strain imposed by the substrate) carried out from first-principles underlines the origin of the different splitting. The model proposed should be valid and applicable to other systems.

Question 2.2 The authors claim that mapping of the Ti3 d orbitals will serve as a stepping stone to understand the microscopic origins of physical phenomena such as chirality and negative permittivity of polar textures. The formation of chiral vortex is usually due to the long-range electrostatic and elastic interactions between different dipoles in the superlattices, in which the change of orbital interactions is not necessary. What's the relationship between the orbital interactions and the chirality of the vortex?

Response:

We agree with the reviewer that formation of chiral vortex is usually due to the long-range electrostatic and elastic interactions between different dipoles in the superlattices. This statement in our paper was simply to point to the microscopic consequences of the interplay between the elastic and electrostatic interactions which lead to the formation of such vortices. Chirality and negative permittivity are physical consequences of such microscopic phenomena. Having said that, we are happy to remove this sentence if the referee so prefers, since it is not central to the main focus of the paper.

In a recent paper, S. W. Lovesey and G. van der Laan [Phys. Rev. B **98**, 155410 (2018); Ref. [19] of the present manuscript], the authors claim that: "The origin of the circular dichroism in Bragg diffraction in PbTiO₃/SrTiO₃ heterostructures is shown by us to be the chiral array of charge quadrupole moments that forms in these heterostructures. While there is no charge quadrupole moment in the spherically symmetric $3d^0$ valence state of Ti⁴⁺, the excited state $2p^53d^1(t_{2g})$ at the Ti L₃ resonance is known to have a quadrupole moment"

The study of the orbital hybridizations and electronic structure of these superlattices is crucial to understand the formation of quadrupole moments, themselves at the basis of the circular dichroism resulting from the chiral structure.

Question 2.3 The detailed information of the first-principles calculations is absent in the manuscript. The periodic boundary conditions are usually employed in first-principles calculations. In the model of Fig.4(a), the boundary conditions are not periodic. How do the authors obtain the vortex of Fig.4(a) from first-principles calculations?

Response:

All details regarding the "description of the methodology can be found in Ref. 34", where this reference points to "Structural and energetic properties of domains in PbTiO₃/SrTiO₃ superlattices from first principles", P. Aguado-Puente and J. Junquera, Phys. Rev. B **85**, 184105 (2012).

In this previous work, some of the authors of the current work provide an exhaustive description of the pseudopotentials, basis set, k-point sampling, construction of the simulation boxes, relaxation thresholds, lattice constants, periodicity, and all the rest of technicalities required to reproduce the calculations of the vortices in $\text{PbTiO}_3/\text{SrTiO}_3$ superlattices. We prefer to cite this reference for the methods employed for the first-principles calculations rather than rewrite them in this paper.

As already well mentioned by the reviewer, we are using periodic boundary conditions along the three directions in real space. However, in the model shown in Fig. 4(a) we only showed part of the supercell used in the simulations. This was done for the sake of clarity in the visualization. Our goal was to focus only on one of the vortices (in this case, the counter-clockwise). In Fig. R5, we provide the whole unit cell (taken from Fig. 3b of the former reference), and the part that was used to make the schematic in Fig. 4(a) of the present manuscript is enclosed in a blue box.

Figure R5. Local polarization profile of polydomain structures in $(\text{PbTiO}_3)_6/(\text{SrTiO}_3)_6$ superlattices. The PbTiO_3 and SrTiO_3 are depicted as gray and white regions respectively. The simulation box enclosed by the solid black line is periodically repeated in space. Taken from Fig. 3(b) of Ref. P. Aguado-Puente and J. Junquera, Phys. Rev. B 85, 184105 (2012). For the sake of clarity in the visualization, only the part of the supercell enclosed in the solid blue box (where we focus only on the counter-clockwise vortex) was used to generate the cartoon in Fig. 4(a) of the present manuscript.

Question 2.4 In the first-principles calculations of Fig.S9, do the authors consider other distortion like oxygen octahedra rotation?

Response:

Thank you for the excellent question. The only distortions considered were related with the ferroelectric mode, and the associated change in the strain field.

The reason for not considering oxygen octahedra rotations is twofold:

While it is true that the coupling between antiferrodistortive (oxygen octahedra rotations) and ferroelectric modes is critically important in the ultrathin limit ($n=1$) - as this coupling gives rise to hybrid improper ferroelectricity [E. Bousquet *et al.* Nature **452**, 732 (2008)], - the coupling constant first decreases in magnitude and then changes sign (from negative to positive) with increasing periodicity. Therefore, for periodicities smaller than $n=3$ the coupling is negative and the two distortions enhance each other, but for $n>3$ the coupling is positive and polar and antiferrodistortive modes become competing distortions (as in bulk). This was discussed in Fig. 4 of a former manuscript written by some of the current authors [“Mixed ferroelectric-antiferrodistortive-strain couplings in monodomain $\text{PbTiO}_3/\text{SrTiO}_3$ superlattices.” P. Aguado-Puente, P. García-Fernández, and J. Junquera, Phys. Rev. Lett. **107**, 217601 (2011)].

On the other hand, the polarization increases with larger periodicity within the PbTiO_3 layer at the center of the domains. Further, the presence of a ferroelectric distortion penalizes the antiferrodistortive modes and the rotation angles become smaller. This was discussed in Fig. 2 and page 3 of the Reference quoted [Phys. Rev. B **85**, 184105 (2012)].

Thus, we consider that the oxygen octahedra rotations will not play a major role in the periodicities studied in the present work.

Reviewer #3 (Remarks to the Author):

The manuscript presents monochromated STEM-EELS mapping at atomic resolution to reveal the evolution of the Ti 3d orbital interactions as a function of their position in a vortex. The theoretically calculated crystal field multiplet EEL spectra are in agreement with experimental EEL spectra and help us to understand the effect at a fundamental level. The research was done in a comprehensive way, and might be suitable for publication after some revision with the following issues.

Response:

We thank the reviewer for appreciating our work. We have addressed most of the comments pointed out by the reviewer.

Question 3.1 Deeper understanding of how the strain and local polarization affect the electronic structure could be obtained from first-principles simulations. The authors carried out density-functional theory (DFT) calculations based on the local-density approximation (LDA) by building $(\text{PbTiO}_3)_6/(\text{SrTiO}_3)_6$ superlattices. The in-plane lattice parameters and unit cell numbers should be provided. From Figure S8, the supercell used could not contain any polar vortex.

Response:

We thank the reviewer for this comment.

All the details regarding the “description of the methodology can be found in Ref. 34”. This reference points to “Structural and energetic properties of domains in

PbTiO₃/SrTiO₃ superlattices from first principles”, P. Aguado-Puente and J. Junquera, Phys. Rev. B 85, 184105 (2012).

In this work, some of the authors of the current work provide an exhaustive description of the pseudopotentials, basis set, k-point sampling, construction of the simulation boxes, relaxation thresholds, lattice constants, periodicity, and all the rest of technicalities required to reproduce the calculations of the vortices in PbTiO₃/SrTiO₃ superlattices. We prefer to cite this reference for the methods employed for the first-principles calculations rather than rewrite them in this paper.

In particular, and in order to answer the questions of the reviewer, the calculations shown in the current work are for a 6/6 periodicity, assuming the theoretical in-plane lattice constant of SrTiO₃.

Indeed, the calculations for Fig. S9 were obtained from a (PbTiO₃)₆/(SrTiO₃)₆ superlattice that contained a polar vortex. But only the layer-by-layer projected density of states corresponding to the center of the up-type domain was shown. The caption states: “Schematic representation of one column of (PbTiO₃)₆/(SrTiO₃)₆ superlattice at the vortex edge (i.e. at the center of a domain) with polarization up.”

In order to clarify this question, we show in Fig. R7 the full superlattice used in the simulation, and the portion of it where the PDOS was computed.

Figure R7. Local polarization profile of polydomain structures in (PbTiO₃)₆/(SrTiO₃)₆ superlattices. The PbTiO₃ and SrTiO₃ are depicted as gray and white regions respectively. The simulation box enclosed by the solid black line is periodically repeated in space. Taken from Fig. 3(b) of Ref. P. Aguado-Puente and J. Junquera, Phys. Rev. B 85, 184105 (2012). The blue rectangle encloses the column at the center of the domain up where the layer by layer PDOS shown in Fig. S9 was computed.

Question 3.2 In Figure 4, no scale bars are given for (b), so it is unclear that whether (a) and (b) have same length scale.

Response:

We thank the reviewer for pointing this out. We have updated the scale bars.

Question 3.3 Note that the authors indicated that one vortex has a dimension of about

10nm in the Introduction. Also note that [(PbTiO3)16/(SrTiO3)16] superlattices (rather than (PbTiO3)6/(SrTiO3)6) are synthesized in experiment.

Upon closer inspection, (a) and (b) show significant difference on the forms of the vortices, see the pattern of arrows and (gray) color codes of (a) and (b). The vortex in (a) seems to be more symmetric than (b), any explanation on this?

Response:

We thank the reviewer for noticing this important detail. From the experimental perspective, Fig. 4b indicates only one specific vortex. The average behavior of the vortices seen in Fig. 1c indicates more or less a symmetric behavior as Figure 4a. Larger scale simulations using second-principles also show more distorted individual vortices.

From a theoretical perspective, we would like to point how the most important features of the vortices characterized experimentally in the 16/16 superlattice are reproduced in our first-principles calculations with a 6/6 periodicity. In particular, the presence of a continuous rotation of the polarization that broadens the domain wall between the up- and down-domains.

Certainly, there are some details that differ. They are related to:

- The periodicity of the supercell: fewer perovskite layers result in a more rapid polarization rotation. For the 6/6 periodicity, this produces a more continuous rotation of the polarization that results in a more symmetric structure. For the 16/16 superlattice the vortices are slightly elongated in a transition from the vortex structure to a more common flux closure domain. From the topological point of view, the two of them are equivalent (same vorticity and chirality).

- Thermal fluctuations: the DFT calculations are carried out at 0 K, and no thermal fluctuations are present, while the experimental characterization has been done at room temperature.

Question 3.4 The authors constructed the corresponding A site displacement vector maps to indicate the location of the vortex cores etc. How about the B site displacement for such vortex determination?

Response:

We have primarily used the strain analysis to locate the position of the vortex structures. We did not show any measurements of the polarization signals (given by the relative shifts of the A and B sites in the perovskite unit cells). Instead, our strain maps were calculated directly from the A lattice positions, with a relatively low resolution (trading some spatial resolution for a smoother, more precise strain value measurement. We can see in this figure from another paper that the A site strain maps are sufficient to determine the vortex positions with high precision: [C Ophus et al., *Ultramicroscopy* 162, 1 (2016)]

[Redacted]

Looking at the images shown in row F of this figure, all components of the strain tensor taken from only the A sites can be used to identify the vortex positions.

Question 3.5 The authors made careful comparison of Ti³⁺ and Ti⁴⁺ reference spectra, indicating absence of the possibility of Ti³⁺ in the system. As both Pb and Sr are bivalent elements, what is the reason to consider the Ti³⁺ state?

Response:

The only reason for us to check for the possibility of Ti^{3+} is prior reports of the formation of Ti^{3+} . (Du et. al. Nature Communications, 10, 4864 (2019). We therefore wanted to eliminate the possibility of Ti^{3+} in our samples. After careful EELS analysis we didn't find any Ti^{3+} in our samples.

Minor points,

Question 3.6 STO adopts a $Pm\bar{3}m$ cubic space group, the symbol of bar should be on top of 3, instead of an underline?

Response:

Thank you for pointing this out. We have corrected this.

Question 3.7 Line 122, ... Ti L edge spectra at the vortex core as compared to Ti^{4+} (ref) indicates a negligible possibility of the presence of Ti^{3+} in our samples. What is the meaning of ref here?

Response:

Ref here means reference. It is referring to the reference spectra for Ti^{4+} . We have put in the full form now to reduce confusion.

REVIEWER COMMENTS

Reviewer #1 (Remarks to the Author):

I find the authors have addressed most of my concerns, so I recommend acceptance of this paper for publication.

Reviewer #2 (Remarks to the Author):

The authors have addressed most of my previous comments. But the authors haven't satisfactorily addressed my following comment "The formation of chiral vortex is usually due to the long-range electrostatic and elastic interactions between different dipoles in the superlattices, in which the change of orbital interactions is not necessary." By using phase field model without quadrupole moment [Nature Communications 12: 2054 (2021); International Journal of Solids and Structures 162: 198-210 (2019)], the polarization vortex (or even chiral vortex) has been successfully reproduced, indicating that the quadrupole moment and orbital interactions play no role in the formation of vortex.

Reviewer #3 (Remarks to the Author):

The authors used energy loss near-edge structure analysis and first principles calculations to study the [(PbTiO₃)₁₆/(SrTiO₃)₁₆] superlattices. They found that the peaks in Ti L-edge EELS shift systematically depending on the direction and magnitude of the local dipole. TEM-EELS mapping at atomic resolution reveals the evolution of the Ti 3d orbital interactions as a function of their position in a vortex. I think this paper can be published in Nature Communication after carefully addressing the previous reviews.

A few minor issues:

If possible, the image resolutions could be enhanced.

In line 80, "+1.20 % on STO, and a +1.3 % strain on PTO". Why the effective digit numbers are not the same between STO and PTO?

More info about the overall shape of polar vortices? Are they an array of tubes?

Can the shape of this polar vortex structures be changed? Can they be controlled by the influence of the external environment, the movement of the vortex, and the change of the vortex direction.

How stable are these vortices?

Reviewer #1: Reviewer #1 (Remarks to the Author):

I find the authors have addressed most of my concerns, so I recommend acceptance of this paper for publication.

Response: We thank the reviewer for the positive review.

Reviewer #2 (Remarks to the Author):

The authors have addressed most of my previous comments. But the authors haven't satisfactorily addressed my following comment "The formation of chiral vortex is usually due to the long-range electrostatic and elastic interactions between different dipoles in the superlattices, in which the change of orbital interactions is not necessary." By using phase field model without quadrupole moment [Nature Communications 12: 2054 (2021); International Journal of Solids and Structures 162: 198-210 (2019)], the polarization vortex (or even chiral vortex) has been successfully reproduced, indicating that the quadrupole moment and orbital interactions play no role in the formation of vortex.

Response:

We thank the reviewer for addressing this point.

We agree with the referee on the fact that the formation of chiral vortex is due to the long-range electrostatic and elastic interaction between the dipoles in the superlattice. With this as the background, we would like to note that while phase field modeling does provide the mesoscale insights into the origins of vortex formation (as described in the original paper by Yadav et al., Nature, 2016), it does not go into the atomic scale electronic structure **consequences of the formation of vortex structures. Thus, our paper brings to focus two issues: (i) the fact that the orbitals in the Ti ion indeed respond as a direct consequence of the competition between the electrostatic and elastic energies; (ii) our first principles modeling provides the atomic scale insights into the consequences of vortex formation since they include all the relevant lattice and electronic degrees of freedom.**

In the phase field model the reviewer is pointing to, the electronic degrees of freedom are integrated out of the simulations and thus, these calculations do not have atomic resolution. The different interactions (including the role of the electrons) are included in the parameters of the model. Therefore, the claims of the reviewer that "the change of orbital interactions is not necessary" and "play no role in the formation of the vortex" are not justified. All the electronic effects are "hidden" below the parametrization of the model. Thus, within the phase field framework, it is not possible to ascertain how important the electronic degrees of freedom are, simply because the electrons are not included in the simulations. Precisely, that is one of the key messages of our paper: for the first time we have carried out accurate first-principles simulations explicitly include the electrons, and check how the polarization vortex alters the local electronic structure, as measured through EEL Spectra. In this sense, we are going one step forward from all other calculations in these polar topological oxide nanostructures.

We hope this clarifies the central message in our paper.

We have now modified the statement related to chirality slightly in the conclusions to make the message more clear to readers:

Finally, mapping of the Ti 3d orbitals will serve as a stepping-stone to understand the microscopic consequences of physical phenomena such as chirality and negative permittivity that have been reported in such polar textures.

Reviewer #3 (Remarks to the Author):

The authors used energy loss near-edge structure analysis and first principles calculations to study the [(PbTiO₃)₁₆/(SrTiO₃)₁₆] superlattices. They found that the peaks in Ti L-edge EELS shift systematically depending on the direction and magnitude of the local dipole. TEM-EELS mapping at atomic resolution reveals the evolution of the Ti 3d orbital interactions as a function of their position in a vortex. I think this paper can be published in Nature Commination after carefully addressing the previous reviews.

A few minor issues:

If possible, the image resolutions could be enhanced.

We have replaced the images with the highest possible resolution images.

In line 80, “+1.20 % on STO, and a +1.3 % strain on PTO”. Why the effective digit numbers are not the same between STO and PTO?

We have corrected the effective numbers.

More info about the overall shape of polar vortices? Are they an array of tubes?

Can the shape of this polar vortex structures be changed? Can they be controlled by the influence of the external environment, the movement of the vortex, and the change of the vortex direction.

How stable are these vortices?

Response: We thank the referee for the positive comments.

We have added a statement about the shape of the vortices.

The referee brings up an interesting question about whether the shape can be changed. Our planar section TEM studies of the vortices (see image below) do show that the vortices are not rigid tubes but are more flexible. We are currently in the process of studying their mobility and deformability with an external in-plane electric field. Since this requires TEM based imaging (due to the length scales involved), these studies will take us at least another 6 months to complete. These results will be reported once these studies are completed.

[Redacted]

REVIEWERS' COMMENTS

Reviewer #2 (Remarks to the Author):

The revised version is publishable.

Reviewer #3 (Remarks to the Author):

I have gone through the responses from the authors, and would like to support accepting the soundness of the research reported. There are indeed more physical phenomena and understanding to investigate on the polar vortex textures, for this submitted paper I have no further questions and concerns.